# Formation of High-Silica Leucocratic Granitoids on the Late Devonian Peraluminous Series of the Russian Altai: Mineralogical, Geochemical, and Isotope Reconstructions

**Nikolay N. Kruk** [1,2], **Olga A. Gavryushkina** [1,2,*] , **Sergey Z. Smirnov** [1], **Elena A. Kruk** [1], **Sergey N. Rudnev** [1] **and Dina V. Semenova** [1]

1   V.S. Sobolev Institute of Geology and Mineralogy Siberian Branch of Russian Academy of Science, 3 Koptyug Ave., 630090 Novosibirsk, Russia; kruk@igm.nsc.ru (N.N.K.); ssmr@igm.nsc.ru (S.Z.S.); krukea@igm.nsc.ru (E.A.K.); rudnev@igm.nsc.ru (S.N.R.); sediva@igm.nsc.ru (D.V.S.)
2   Geology-Geophysical Department, Novosibirsk State University, 1 Pirogova Str., 630090 Novosibirsk, Russia
*   Correspondence: o.gavriushkina@g.nsu.ru

**Abstract:** This paper presents data on the geological position, geochemical features, main mineral composition (micas, feldspars), and melt and fluid inclusions in quartz from Aba high-silica leucocratic granitoids in the western part of the Talitsa batholith, Russian Altai. According to these new geochemical data, the granitoids are classified as S-type, meaning they are formed via the partial melting of metasedimentary source rocks. Geological data and oxygen isotope composition analysis indicate that major-phase granitoid magma evolution took place at the level of intrusion formation, whereas the parent melt of late-phase leucogranite evolved in a deeper chamber. The geochemical features (HFSE and REE, and REE spectra) of the granitoids indicate significantly higher differentiation in the late leucocratic phase. The presence of coexisting syngenetic melt and fluid inclusions shows that leucogranite magma was already saturated with volatiles in the early crystallization stages. Based on the new data presented in this work, the Aba rock formation is associated with the volatile saturation of magmatic melts, the exsolution of a fluid phase, and magma degassing.

**Keywords:** leucogranite; granitoids magmatism; mineralogy; geochemistry; Altai





## 1. Introduction

The formation of high-silica leucocratic granites is one of the most poorly understood processes in the petrology of silicic rocks. Leucogranites are often associated with the development of various intrusive series, and the presence of individual intrusions is uncommon. The formation of leucogranite melts is most commonly associated with the deep differentiation of granitoid magmas [1] or with low-degree melting of the continental crust [2]. In this regard, the formation mechanism of high-silica leucogranites with excess quartz in comparison with eutectic granite composition is as yet unsolved. Such melts may originate during the evolution of less silicic magmas in a closed system. It was previously assumed [3] that leucocratic granitoids with $SiO_2$ concentrations of up to 76 wt.% and without any inclusions are pure anatectic melts that are not contaminated by restite entrainment. Subsequently, numerous experiments on the melting of various crustal rocks disproved this point of view. It was found that melts with an excess of normative silica are formed during the melting of specific silica-enriched rocks that are enriched in potassium and volatile components [4]. However, such rocks are rare in the Earth's crust and cannot explain the wide distribution of high-silica leucogranites. It should also be noted that different series of leucogranites have similar mineral and chemical compositions and are indistinguishable on most compositional diagrams (for example, [5,6]), suggesting that specific petrogenetic mechanisms are involved in the formation of high-silica leucogranite magmas.

The interaction of a near-eutectic melt with a fluid that is enriched in silica, alkali elements, and a number of rare elements may be one such mechanism. The frequently observed tetrad effect in leucogranites confirms such a mechanism [7–9]. This phenomenon, which affects chondrite-normalized rare Earth element (REE) spectra, can be explained by differences in the distribution coefficients of lanthanides between coexisting fluids and melts, and is the result of the interaction of the melt with a separate fluid phase. Experimental studies (such as [10]) have also confirmed the possibility of high-silica magma formation occurring due to the introduction of silica through a fluid phase. In these experiments, the interaction of metabasalts with a $SiO_2$-rich fluid resulted in 'granite-like' rocks with $SiO_2 > 60$ wt.%. At the same time, several details remain unclear. Firstly, the composition of the fluid after its interaction with metabasalts was not studied in previous experiments; thus, it is impossible to estimate the extent of the removal of components from the system during the interaction. Secondly, it is not clear whether silica abundance depends upon $SiO_2$-rich fluids in silicic magmas of granitic composition. It should be noted that there are few detailed descriptions of the geological materials formed by the process described above. In this study, the formation of high-silica leucocratic granitoids was modeled using rocks of the Late Devonian high-alumina associations of Gorny Altai (Russia).

## 2. Geological Background

Gorny Altai is a region that has undergone prolonged, multi-stage geological formation and evolution. Details of the geological structure and tectonics of this region are given in [11–23]. The main volumes of the granitoids of Gorny Altai originate from the Late Devonian and Permian–Triassic eras [24–27] (Figure 1). Devonian granitoids were formed along the active margin of the Siberian continent at the boundary with the Ob'–Zaisan oceanic basin [15,16,25]. Permian–Triassic granitoids are products of intraplate magmatism [24–26,28,29].

The change in the continental margin's geodynamic setting from subduction- to transform-type tectonism caused a surge in Late Devonian granitoid magmatism. At that time, suprasubduction volcanism along the linear magmatic belts was replaced by the formation of local volcanic areas with alkaline bimodal series, including OIB-like basalts [30], small hypabyssal gabbro–granite and granitoid intrusion formation, and then, the intrusion of large granitoid batholiths; the largest of these is the Talitsa batholith in the western part of Gorny Altai (Figure 2).

The Talitsa batholith contains two main rock associations: one composed mainly of gabbro, metaluminous quartz diorites, granodiorites, and granites, and a second late association that includes granitoids of the peraluminous series. The latter are widely developed in the western part of the Talitsa batholith, south of the village of Charyshskoye. There are several individual intrusions and many small massifs composed of granodiorites, and melano-, meso-, and leucocratic granites that contain red-brown aluminous biotite, muscovite, and, in some cases, cordierite that intrudes biotite–amphibole rocks of the first association. One of the individual intrusions is the Charysh massif, which is composed of porphyritic biotite (rarely with hornblende) granodiorites and melanocratic granites (Figure 2). In the eastern part, this massif intrudes biotite–amphibole granodiorites, whereas in its western part, it intrudes Early Paleozoic terrigenous–volcanogenic–siliceous sediments. To the north, there is the large Borovlyanka intrusion, composed mainly of porphyritic biotite melanocratic granites (Figure 2). In the southern part, the rocks of the Borovlyanka massif are intruded by granite–leucogranites of the Aba intrusion.

The Aba massif has an irregular shape, an area of about 50 km$^2$, and is elongated in a submeredional direction. In the eastern and southern parts of the massif, the rocks intrude weakly metamorphosed Late Cambrian–Early Ordovician terrigenous deposits, and in the west, they intrude melanocratic granites of the Borovlyanka intrusion. The contacts are intrusive, and well-marked by changes in melanocraticity and rock structure.

The Aba massif is composed of rocks of two intrusive phases—coarse-grained porphyritic biotite granites and leucogranites of major-phase rocks—with a gradual transition between them. In the southern part of the massif, the rocks of the main phase are intruded by two-mica and muscovite leucogranites. These rocks form separate small outcrops among the Quaternary formations. Leucogranites are fine- and medium-grained, generally equigranular, and rarely contain quartz–feldspar pegmatoid segregations of less than 10 cm in size. Studies of petrographic thin sections allowed us to identify traces of brittle-plastic deformation in leucogranites of the late phase. Moreover, the presence of gneiss-like and cataclastic textures in these rocks enabled us to confirm deformation processes. The gneissic layering orientation is generally close to vertical, but tends to change across the strike and varies even within small outcrops.

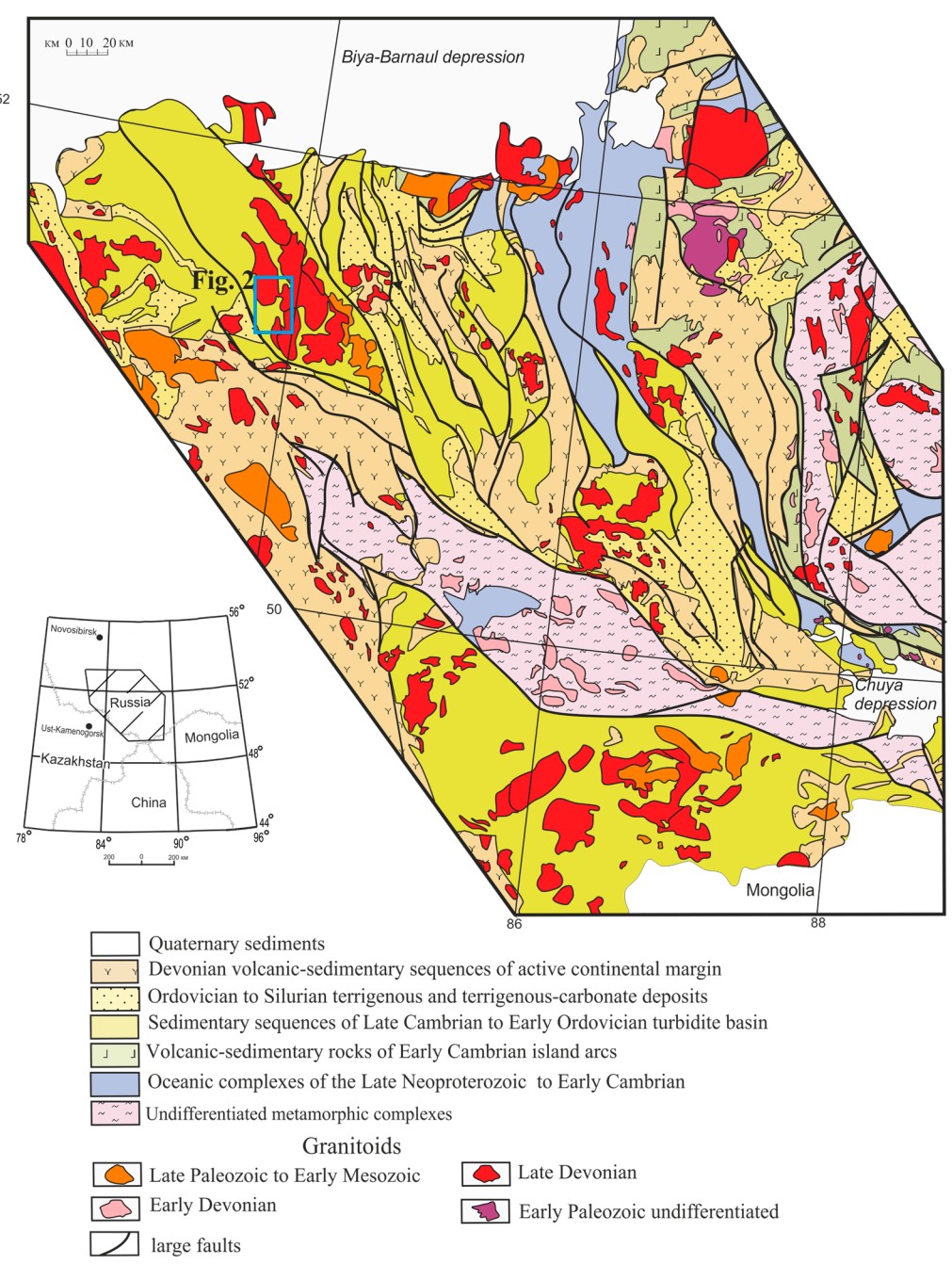

**Figure 1.** Generalized geology of the Gorny Altai after [27].

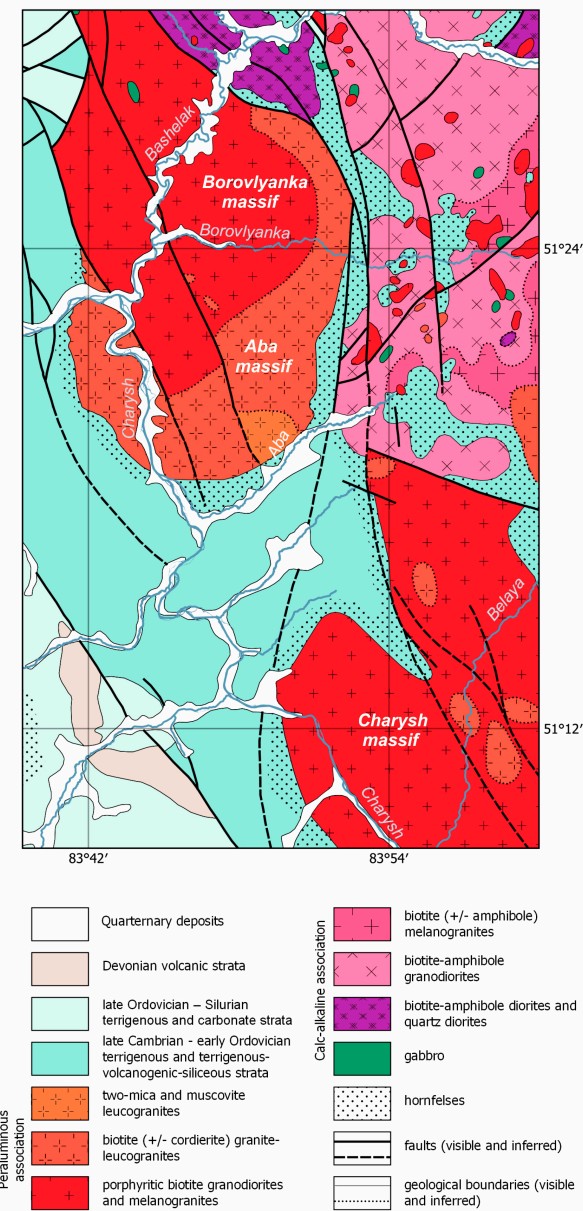

**Figure 2.** Geological scheme of the western part of Talitsa batholith after [25] with additions.

In addition, the granite–leucogranites of the major-phase rocks are intruded by rare dikes of muscovite aplites, with thickness ranging from 0.4 to 1.5 m.

## 3. Analytical Methods

### 3.1. U–Pb Geochronology

The U–Pb ages of single zircon grains were measured via LA-SF-ICP-MS using a Thermo Fisher Scientific Element XR magnetic sector-field ICP mass spectrometer, coupled with a New Wave Research UP-213 Nd: YAG laser ablation system, at the Analytical Center for Multi-Elemental and Isotope Research (IGM SB RAS, Novosibirsk) [31]. The grains were mounted in epoxy resin, together with Plešovice zircon standards. The analyzed zircons were polished to about one half of their thickness. Spots for laser ablation were chosen using optical and catholuminescence (CL) images. All measurements were performed using electrostatic scanning (E-scan) at masses of $^{202}$Hg, $^{204}$(Hg + Pb), $^{206}$Pb, $^{207}$Pb, $^{208}$Pb, $^{232}$Th, and $^{238}$U. The signals were detected in counting mode for all isotopes, except for $^{238}$U and $^{232}$Th, for which triple mode was applied. The zircons were analyzed using a laser spot of 25–30 μm in diameter, with a fluence of 2–3 J/cm$^2$ and a repetition rate of 10 Hz, for

30 s. Data reduction was carried out using the Glitter software package [32]. Concordia ages and diagrams were generated using the Isoplot-3 software package [33].

### 3.2. Whole-Rock Geochemical Analyses

The studied samples were selected after excluding those that were weathered or subjected to postmagmatic changes.

Major and trace elements were analyzed at the Sobolev V.S. Institute of Geology and Mineralogy at the Siberian Branch of the Russian Academy of Sciences (IGM SB RAS, Novosibirsk, Russia) at the Shared-Use Analytical Center for Multielement and Isotopic Studies. The concentrations of major elements were measured via X-ray fluorescence (XRF) using an ARL-9900-XP spectrometer (Applied Research Laboratories) according to the standard procedure. Rare elements were analyzed using a Finnigan Element inductively coupled plasma mass spectrometer (ICP-MS) following the protocol of [34].

Measurements of fluoride ions were performed via potentiometry at the Institute of Geochemistry (Shared-Use Analytical Center for Isotopic and Geochemistry Studies) at the Siberian Branch of the Russian Academy of Sciences, Irkutsk, Russia. A fluoride electrode (ELIT-221) and a silver chloride reference electrode (EVL-1M3.1) were used in the current study. Distilled water and reagents were used to prepare the following solutions: $CH_3COONa$, $Na_2B_4O_7 \cdot 10H_2O$ (borax), $Na_2CO_3$ (soda), $CH_3COOH$, $Na_3C_6H_5O_7$, NaF, KCl, and HCl. Before measurements were performed, the analyzed rock samples (0.1 g sample) were transferred into the solution via fusion with a mixture of soda and borax (2:1). In the next step, we performed dissolution of the melt in a solution of hydrochloric acid. The pH of the solutions was maintained using an acetate buffer solution consisting of sodium acetate and glacial acetic acid ($5.5 \pm 0.1$). The buffer was added to the examined solution in a 1:1 ratio. To eliminate the overlap of several elements (calcium, aluminum, barium, magnesium, and strontium) in the determination of fluorine, sodium citrate solution (1 M) was introduced into the analyzed solutions. Measurements were performed in solutions of standard samples and analyzed samples. The fluoride ion was calculated using calibration dependence (E vs. pF).

### 3.3. Whole-Rock Isotopic Analyses

Sm–Nd isotopic studies were performed at the Geological Institute of Kola Scientific Center, Apatity city (Russia) using a Finnigan MAT 261 8-collector mass spectrometer in static mode. Rock powders for the Sm–Nd studies were dissolved in a mixture of HF, $HNO_3$, and $HClO_4$. Before the decomposition occurred, all samples were completely spiked with a $^{149}Sm$–$^{150}Nd$ mixed solution. The REEs were separated using conventional cation-exchange techniques. Sm and Nd were separated via extraction chromatography using Eichrom LN-Specresin columns. The total blanks in the laboratory were 0.1–0.2 ng for Sm and 0.1–0.5 ng for Nd. The accuracy of the measurements of the Sm and Nd content was $\pm 0.5\%$, $^{147}Sm/^{144}Nd$—$\pm 0.5\%$, and $^{143}Nd/^{144}Nd$—$\pm 0.005\%$ (2r). $^{143}Nd/^{144}Nd$ ratios were normalized to a value of 0.511860 for the La Jolla standard. During the experimental period, the weighted average of nine La Jolla Nd-standard runs yielded $0.511852 \pm 8$ (2r) for $^{143}Nd/^{144}Nd$, using 0.7219 to normalize $^{146}Nd/^{144}Nd$. The $\varepsilon Nd_{(T)}$ values were calculated using the present-day values for a chondritic uniform reservoir (CHUR) ($^{143}Nd/^{144}Nd = 0.512638$ and $^{147}Sm/^{144}Nd = 0.1967$) [35]. The model ages (TDM) were calculated using a model [36], according to which the Nd isotope composition of the depleted mantle has evolved linearly since 4.56 Ga ago, and has a present-day $\varepsilon Nd_{(0)}$ value of +10 ($^{143}Nd/^{144}Nd = 0.513151$ and $^{147}Sm/^{144}Nd = 0.2137$). The two-stage Nd model ages (TDM2) [37] were calculated using the crustal mean ratio $^{147}Sm/^{144}Nd = 0.12$ [38].

### 3.4. Mineral Analyses

The chemical composition of rock-forming minerals was determined at the Sobolev V.S. Institute of Geology and Mineralogy at the Siberian Branch of the Russian Academy of Sciences (Shared-Use Analytical Center for Multielement and Isotopic Studies), Novosibirsk,

Russia, via electron microprobe analysis (EMPA) using a JXA-8100 (JEOL) microanalyzer with five wavelength-dispersive spectrometers. The analyses were carried out at an accelerating voltage of 20 kV, with a beam current of about 30 nA, and an electron beam diameter of 2 μm. A set of well-characterized intralaboratory standards (albite (Na, Al), orthoclase (K), fluoro-phlogopite (F), and diopside (Ca, Mg, Si)) were used to calibrate the instruments. The following are the limits of detection for impurity elements, $3\sigma$ (wt.%): FeO 0.05, MnO 0.07, BaO 0.24, $Na_2O$ 0.14, MgO 0.07, $Rb_2O$ 0.27, $K_2O$ 0.03, CaO 0.02, $TiO_2$ 0.07, $P_2O_5$ 0.13, $Cs_2O$ 0.06, F 0.2, and Cl 0.01. Accessory minerals were analyzed via energy-dispersive X-ray spectroscopy (EDS) using a Tescan MIRA 3 LMU, coupled with energy-dispersive spectroscopy. For EDS analyses, the beam current was 1 nA, the beam diameter was 10 nm, and the analysis was carried out by scanning an area of $5 \times 5$ μm. The live spectrum acquisition time was 60 s. The stability of the survey parameters was controlled by periodically measuring the intensity of the $K_\alpha$ line of pure cobalt. The correctness of the obtained results was controlled via periodic measurement of the standards used in the calibration.

The isotope composition of oxygen in minerals (quartz) was defined using a Finnigan MAT 253 gas mass spectrometer with a classic-variant double system of inflow (standard-sample) at the Dobretsov Geological Institute at the Siberian Branch of the Russian Academy of Sciences (Shared-Use Analytical Center "Geospectr"), Ulan-Ude, Russia. To determine the $\delta^{18}O$ values, samples were prepared via laser fluorination (LF) in the presence of $BrF_5$ reagent [39]. The array involved the MIR device's 10–30 heating system, and included laser $CO_2$ with a capacity of 100 Wt and a wavelength of 10.6 lm in the infrared area, which allowed for heating of the analyzed minerals to 1000 °C.

The resulting reaction was visually monitored to observe the completeness of decomposition. At times, we had to change the regime (the output and the focus of the laser beam) to achieve complete sample combustion. In LF, oxygen is not fractionated because of the short reaction time and high temperature. The decomposition of one sample requires about 15–20 min, minimizing contamination of the obtained gas by atmospheric impurities.

Only fragments of pure minerals weighing 1.5–2.5 mg were used for the isotope analysis. The $\delta^{18}O$ values were calculated using the international standards NBS-28 (quartz) and NBS-30 (biotite), and checked using the internal standard GI-1 (quartz) and one Polaris (quartz) of IGEM RAS. The error of the estimated $\delta^{18}O$ values was (1s) $\pm$ 0.2%.

## 4. Results

### 4.1. Petrography

The granodiorites and melanogranites of the Charysh and Borovlyanka massifs undergo gradual transitions, have similar structural and textural features, and differ only in the quantitative ratios of their rock-forming minerals. The rocks are characterized by the presence of porphyritic phenocrysts of gray K-feldspar (1–5 cm) and white plagioclase (1–2 cm). Phenocrysts occupy 40% of the rock volume in granodiorites, and up to 15% in melanogranites. Sometimes, large phenocrysts are absent but the texture of the rocks is still unevenly grained. K-feldspar phenocrysts contain inclusions of biotite and plagioclase. The groundmass has a hypidiomorphic granular microtexture due to the idiomorphism of plagioclase. Hypidiomorphic and serial-porphyritic textures are common. There are medium-sized (up to 15 cm) melanocratic rounded inclusions that are composed of fine-grained plagioclase–biotite–quartz rocks and are sometimes porphyritic. Rarely, the rocks contain muscovite and show evidence of cataclasis.

Granite–leucogranites of the major phase of the Aba massif are predominantly light-gray-to-gray coarse- and medium-grained rocks. The rocks are composed of smoky quartz (30%–38%), plagioclase (25%–30%), latticed microcline–perthite (30%–35%), and red-brown biotite (4–10%) (Figure 3a). The presence of dark gray columnar crystals of pinitized cordierite measuring up to 2 cm long is a special feature of these granite–leucogranites. Often, there is an admixture of muscovite.

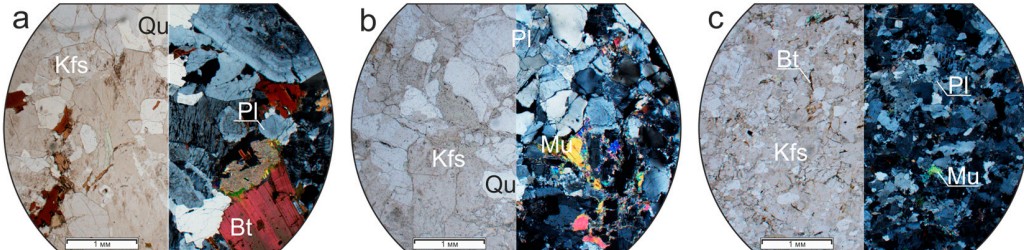

**Figure 3.** Petrography of the Aba rocks. Right and left panels are photographs in polarized and transmitted light, respectively. (**a**) granite of the major phase; (**b**) muscovite leucogranite of the late phase; (**c**) muscovite aplite of the dike.

The field aspects of these rocks depend on the amounts (from 20% to complete absence) of elongated gray K-feldspar phenocrysts they contain, which measure up to 5 cm in size. Their groundmass is medium–coarse-grained, formed of small isometric grains of perthite, latticed microcline, tabular segregations of acid plagioclase, round grains of 'blocky' quartz, and flakes of reddish-brown biotite ($f$ = 62%–73%), and is idiomorphic in relation to other minerals of the hypidiomorphic groundmass. A decrease in the number of phenocrysts in their groundmass leads to an increase in K-feldspar and quartz, and a decrease in biotite. In general, the granite varieties are porphyritic, and leucogranites do not have phenocrysts.

Two-mica and muscovite leucogranites of the late phase are uneven- and fine–medium-grained rocks with a grain size of 0.5 to 2.5 mm. These rocks are composed of quartz (35%–40%), plagioclase (25%–40%), K-feldspar (25%–35%), muscovite (2%–5%), and biotite (0%–2%) (Figure 3b). Plagioclase shows characteristic polysynthetic twinning, and microcline displays gridiron twinning. The texture of the rocks is hypidiomorphic-granular, but close to the fault zones, it is usually cataclastic and contains small angular quartz grains with a cloudy extinction.

Dikes of muscovite aplites are composed of white or light-gray fine-grained rocks, consisting of quartz (40%–45%), plagioclase (20%–25%), K-feldspar (25%–30%), and fine-flake muscovite (Figure 3c). Phenocrysts are absent, and grain size does not exceed 2 mm. The structure of the rocks is aplite, and sometimes cataclastic.

Accessory minerals are generally unvaried. They include magnetite, ilmenite, apatite, zircon, monazite, and, less often, orthite and tourmaline. Granite–leucogranites of the major phase of the Aba massif also contain xenotime and cheralite. Usually, accessory minerals form euhedral inclusions in rock-forming biotite, plagioclase, and quartz. Apatite is present either as small elongated euhedral crystals (up to 50–150 μm) and large xenomorphic grains (80–300 μm) in the interstices between rock-forming biotite and feldspars, and often carry inclusions of other accessory minerals. Ilmenite, like apatite, occurs both as small euhedral crystalline inclusions (40–80 μm) in biotite and quartz, or as grains without crystallographic outlines (in this case, the grain size of ilmenite varies from 100 to 300 μm, occasionally reaching 500 μm). At the same time, xenomorphic grains are found only in biotite, and often repeat the outlines of its crystals.

*4.2. Geochronology*

The U–Pb dating of zircon grains extracted from rocks of the Charysh, Borovlyanka, and Aba massifs provided a time range for the emplacement of the peraluminous granites of the western part of the Talitsa batholith. Zircons of the Charysh massif are derived from medium-grained porphyritic melanocratic biotite granites (sample 12-056). Zircon grains are transparent to translucent, colorless, or predominantly pale pink euhedral prismatic crystals with smooth edges and prismatic faces measuring 20 to 250 μm long (Kelong = 2.5–5.0). A total of 23 point analyses were conducted. Isotopic $^{206}Pb/^{238}U$ ratios from eighteen analyses of zircons yielded an age range of 378–385 Ma (Table S1), with a weighted average age of 380 ± 5 Ma (MSWD = 0.036). The Th/U ratios of the studied zircons range from 0.08 to 0.75, which overlap with the compositional ranges of magmatic

zircon. The concordant age for these grains (Figure 4a) is $379 \pm 3.4$ Ma (MSWD = 0.13). The other five zircon grains, with concordant ages of 393–394, $426 \pm 6$, $474 \pm 7$, and $56 \pm 7$ Ma, are independent clusters and demonstrate a xenogenic nature.

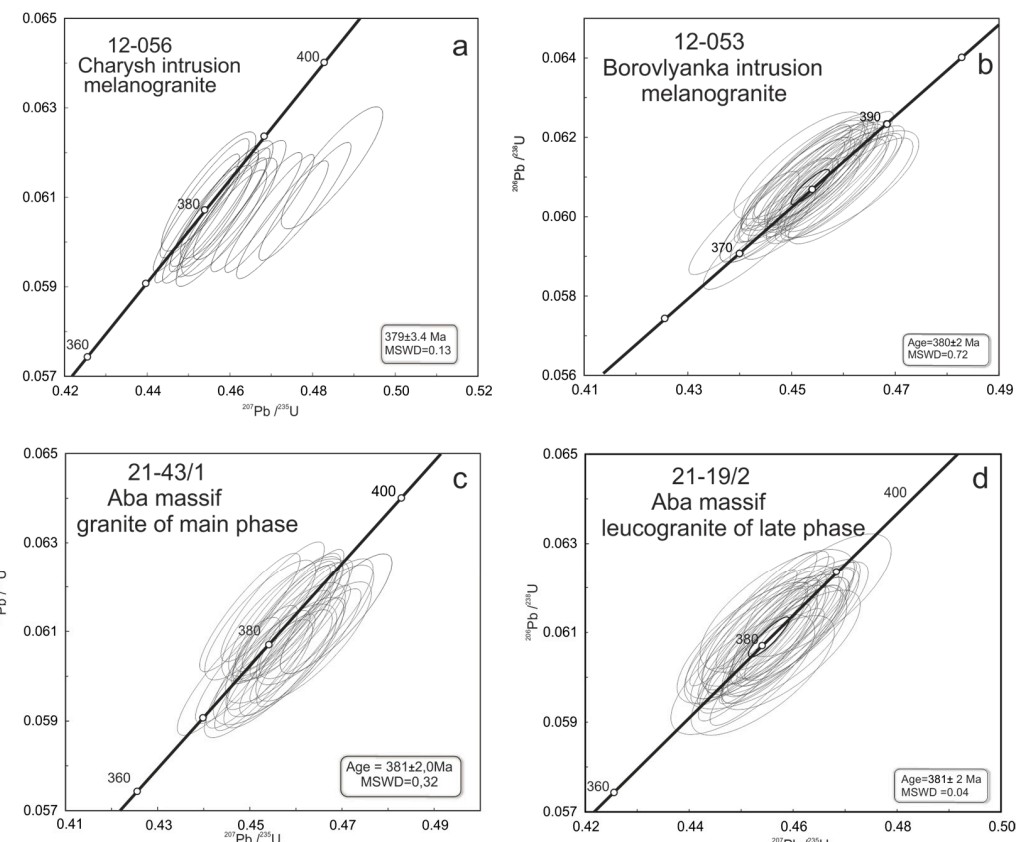

**Figure 4.** U–Pb isotopic diagrams for Late Devonian granitoids of the western part of the Talitsa batholith (Gorny Altai).

Zircons extracted from coarse-grained porphyritic melanocratic granites of the Borovlyanka massif (sample 12-053) contain pale yellow and dark pink (predominant) to colorless transparent and translucent euhedral crystals. The zircon grain sizes vary from 20 to 160 μm, and in some cases, reach 250 μm. The crystals have a generally prismatic habit, with $K_{elong}$ = 2.5–5.0; long prismatic crystals ($K_{elong}$ = 7–10) are very rare. The CL images show magmatic oscillatory zoning; sometimes, zircons contain relics of 'ancient' cores. A total of 36 analyses were carried out. The zircon Th/U ratios range from 0.08 to 0.34, and sometimes to 0.58 (Table S1); this, together with the CL studies, indicates their magmatic origin. The $^{206}Pb/^{238}U$ ratios from 35 analyses yielded an age distribution from 328 Ma to 383 Ma (Table S1), with a weighted average age of $380 \pm 5$ Ma (MSWD = 0.065). The concordant age of these grains (Figure 4b) is $380 \pm 2$ Ma (MSWD = 0.72). One grain shows a concordant age of $395 \pm 3$ Ma.

The Aba massif samples were taken from a coarse-grained biotite porphyritic leucogranite of the first phase (sample 21-43/1) and a medium-grained two-mica leucogranite of the late phase (sample 21-19/2). The zircon fraction from granites of the major phase (sample 21-43/1) is represented by colorless, pale yellow to pale pink transparent (dominantly) and translucent idiomorphic crystals with smooth edges and faces. The grain size of zircons varies from 50 to 250 μ. The crystals have a generally prismatic habit, with $K_{elong}$ = 2.5–4.5; long-prismatic crystals ($K_{elong}$ = 7–10) are very rare. The CL images show characteristic magmatic zoning. Of 33 individual analyses (Table S1), 29 yielded an age distribution of 375–386 Ma, with a weighted average age of $380 \pm 4$ Ma (MSWD = 0.10). The Th/U ratios range from 0.07 to 0.58, which is typical for magmatic zircons. The concordant age

(Figure 4c) is 381 ± 2 Ma (MSWD = 0.32). The ages of the other zircons are as follows: three points show an age range of 393–396 Ma, the weighted average age is 395 ± 12 Ma (MSWD = 0.02), and for one local point, the age of zircon is 419 ± 6 Ma.

Zircons from leucogranite of the late phase of the Aba massif (sample 21-19/2) do not differ in their morphological characteristics, dimensions, and color from zircons from granites of the major phase of the same massif, and also have magmatic zoning.

A total of 40 point analyses of zircon were carried out to enable their identification (Table S1). The isotopic ratios (for $^{206}Pb/^{238}U$) of 34 local points yielded ages between 376 and 386 Ma (Table S1), with a weighted average age of 381 ± 4 Ma (MSWD = 0.06) and a concordant age (Figure 4d) of 380 ± 2 Ma (MSWD = 0.04). The Th/U ratios of the studied zircons range from 0.04 to 0.41, which overlap with the compositional ranges of magmatic zircon. The ages of the other six zircon grains are as follows: five points show an age range of 390–398 Ma, the weighted average age is 396 ± 10 Ma (MSWD = 0.11), the concordant age is 396 ± 5 Ma (MSWD = 0.10), and one zircon crystal yielded an age of 478 ± 7 Ma.

In summary, the ages obtained for the studied massifs of peraluminous granitoids in the western part of the Talitsa batholith coincide within the realm of analytical error. The formation and emplacement of these massifs occurred during one stage of magmatic activity within a narrow time interval (380 ± 5 Ma).

### 4.3. Geochemistry and Isotope Characteristics of Rocks

The granitoids described above share similar lithogeochemical features. They belong to the calc-alkaline series (Figure 5a), and most of them are characterized by $K_2O/Na_2O > 1$, correspond to high-potassium rocks (according to [40]) (Figure 5b), vary from calc-alkaline to alkali-calcic on the "$SiO_2$-MALI" diagram (Figure 5e), and have elevated phosphorus content (Table S2).

These granitoids are supersaturated in alumina; A/CNK changes gradually from 1.03 to 1.06 for the granodiorites of the Charysh intrusion to 1.23–1.26 for the late-phase two-mica and muscovite leucogranites of the Aba intrusion (Figure 5c). The melanocratic granites of the Charysh and Borovlyanka intrusions are magnesian; the granites and leucogranites of the Aba massif are mostly ferroan (Figure 5d).

The melanocratic granites of the Charysh and Borovlyanka intrusions are quite similar and do not show substantial variation in their composition. The rocks contain $SiO_2$ at 65–70 wt.%, $Al_2O_3$ at 14–16.3 wt.%, low CaO content and femic elements, and are relatively enriched in phosphorus (up to 0.7 wt.% $P_2O_5$ in some samples). The amount of normative corundum does not exceed 2 wt.%.

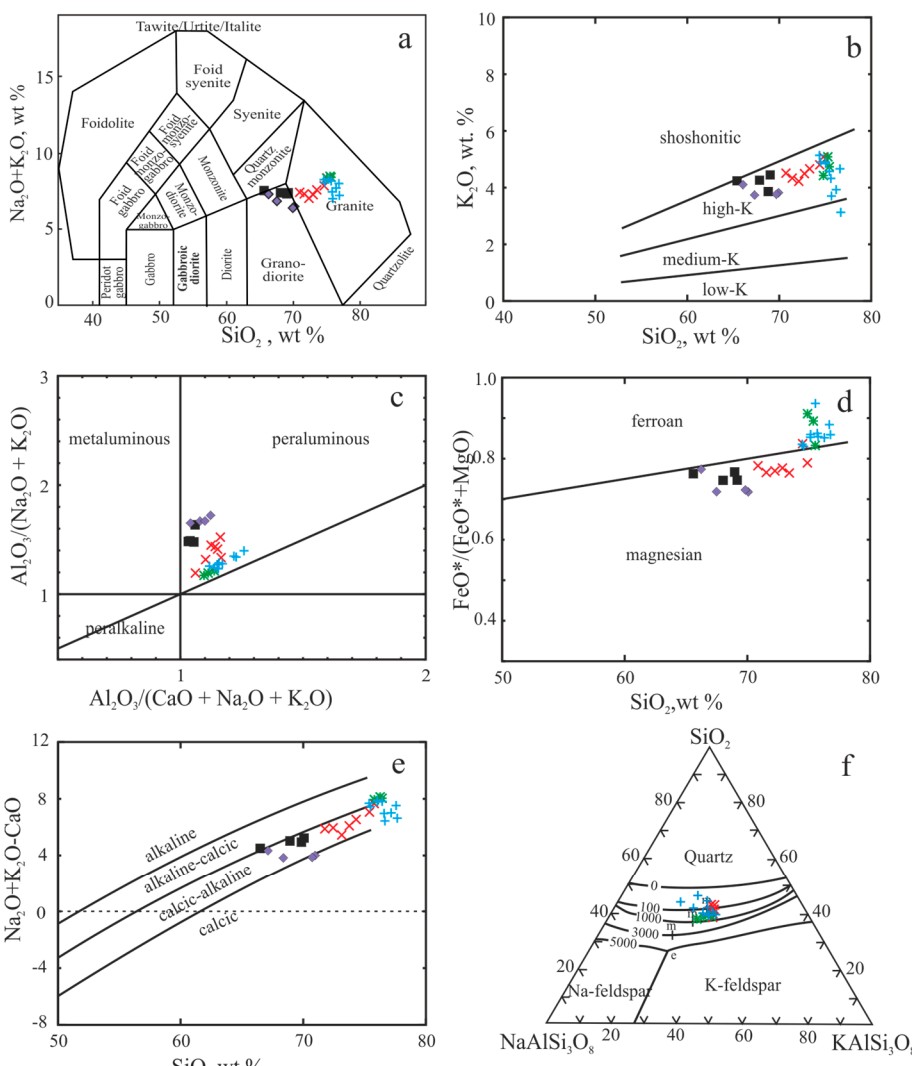

**Figure 5.** Major-element diagrams for Late Devonian granitoids of the western part of the Talitsa batholith (Gorny Altai). (**a**) TAS diagram, boundaries based on [41]; (**b**) SiO$_2$ vs. K$_2$O diagram, boundaries based on [40]; (**c**) ACNK vs. ANK diagram, based on [42], (**d**) SiO$_2$ vs. FeO*/(MgO) + FeO* diagram [6]; (**e**) SiO$_2$ vs. MALI diagram [6]; (**f**) quartz–albite–orthoclase diagram. Solid lines—cotectics, numbers represent pressure (bar); M—isobaric minima; e—point of composition of granite that is eutectic at P = 5 kBar.

The concentrations of most of the incompatible elements are close to those described for low-Rb S-type granitoids (according to [43]). The average upper continental crust concentrations of the REEs of the least silicic rocks are 196 and 158 ppm for the Charysh and Borovlyanka intrusions, respectively. The REE spectra display a fractionated pattern of (La/Yb)$_N$ = 4.7–5.9 and 4.9–7.7 for the Charysh and Borovlyanka intrusions, respectively (Figure 6). A small negative Eu-anomaly is common for samples of both the investigated intrusions. In granitoids of the Borovlyanka massif, the amounts of REEs and the degree of differentiation in their spectra slightly decrease, and negative Eu-anomalies become

slightly deeper as silica content increases. Multi-element diagrams display enrichment in Ba and depletions of Ta, Nb, and Ti (Figure 6). The concentration of fluorine in granitoids does not exceed 0.05 wt.% and increases with increasing silicic acidity.

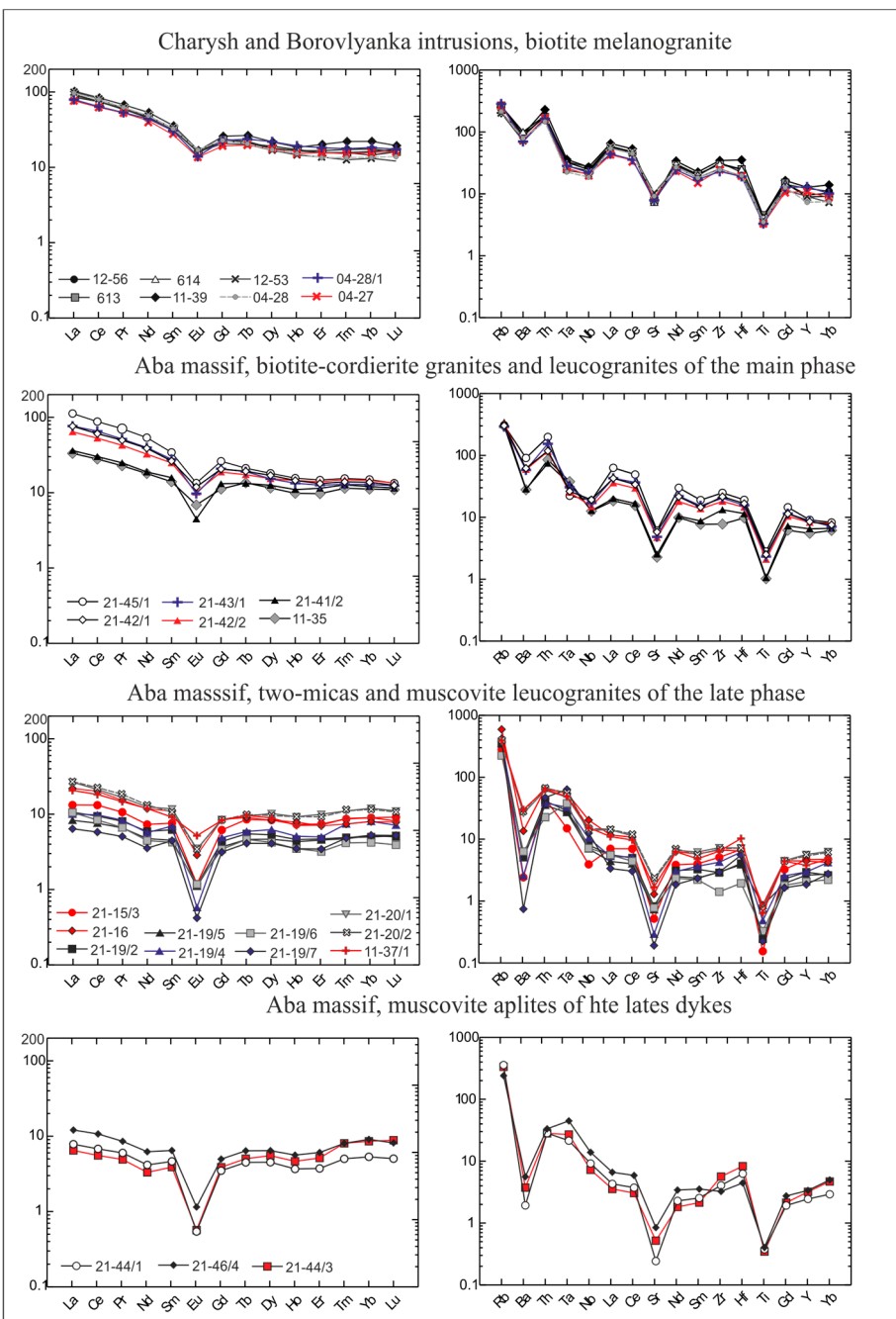

**Figure 6.** REE patterns and spidergrams for Late Devonian granitoids of the western part of the Talitsa batholith (Gorny Altai). Sample numbers are the same as in Table S2. REE spectra are normalized to chondrite based on [44], and spidergrams are normalized to primitive mantle based on [38].

The granite and leucogranite of the Aba intrusion have wider variations in composition. Granite–leucogranite of the major phase contain SiO$_2$ at 70–74 wt.%, and two-mica and muscovite leucogranite contain SiO$_2$ at 74.5–76.5 wt.%. The rocks are characterized by a low content of femic components and calcium, the concentrations of which decrease with increasing silicic acidity. The least siliceous biotite porphyritic granites (samples 45/1 and 42/1, SiO$_2$ content of 70–71 wt.%) are similar to the melanogranite of the Borovlyanka in-

trusion in terms of their major element content and their REE distribution spectra (Table S2, Figure 6). With an increase in silica (samples 41/2 and 35, in which the content of $SiO_2$ is more than 74 wt.%), the alkalinity, iron, and alumina indexes of rocks increase (*f*—up to 84%, A/CNK—up to 1.17). At the same time, an increase in silica does not influence the total alkali content or the $K_2O/Na_2O$ ratios. In the "Qz-Ab-Or" diagram (Figure 5f), the composition of granite–leucogranite in the major phase corresponds to the area of the surface granite cotectic. The most siliceous compositions are shifted to the field of quartz crystallization.

In the more silica-rich leucogranite, there is a tendency for REE concentration to decrease from the least silicic varieties to more silicic ones (from 247 ppm to 137 ppm); moreover, there is a decrease in the degree of REE differentiation (La/YbN from 7.5 to 3.0–3.6), and the negative Eu-anomaly becomes slightly more pronounced (Figure 6). The REE distribution spectra of the more silicic biotite leucogranite, 11–35, shows a pronounced M-type tetrad effect (T = 1.11). The concentrations of the most highly charged elements (HFSE) also decrease, except Ta and Nb; their concentrations do not change and, as a result, they do not have negative anomalies on the spidergrams (Figure 6). The content of fluorine (F-0.07 wt.%) in the least silicic samples of biotite granite–leucogranite is slightly higher compared to the granitoids of the Charysh and Borovlyanka intrusions. With an increase in silica, the F-concentration first slightly increases (up to 0.08 wt.% at $SiO_2$ = 72–73 wt.%), and then decreases to 0.06 wt.%. Concentrations of large-ion lithophile elements (LILEs) vary widely, are practically independent of increases in the silica content of the rocks, and Sr and Ba content decreases (Figure 7).

Two-mica and muscovite leucogranites of the late phase differ from biotite granite–leucogranites in that they have higher silicic acidity, a higher iron index, and lower femic element and calcium contents. At the same time, in the least silicic two-mica leucogranites, the total alkalinity and the ratio of potassium to sodium are the same as in the biotite leucogranites described above ($K_2O + Na_2O$= 8.0–8.2 wt.%, $K_2O/Na_2O$ = 1.57–1.67). Further, with an increase in silica, total alkalinity decreases to 7%, and the $K_2O/Na_2O$ ratio in the more silicic samples varies from 1.4 to 0.7. The values of the A/CNK index in two-mica and muscovite leucogranites are much higher than in the previous granitoids (1.12–1.26), and increase slightly with increasing $SiO_2$. In the "Qz-Ab-Or" diagram (Figure 5f), the composition points of two-mica and muscovite leucogranites of the late phase of the Aba massif form an elongated distribution from the cotectic minimum to the quartz crystallization field. The most siliceous compositions are shifted to the Qz-Ab side of the composition (Figure 5f).

The least silicic two-mica leucogranites, in comparison to biotite leucogranites of the major phase, are characterized by higher LILE contents and lower concentrations of Sr, Ba, and F (0.05 wt.%). With an increase in silica in leucogranites, we observe a decrease in total alkalinity, mainly due to potassium; in more silicic samples, $K_2O/Na_2O$ = 1.4–0.79. Thus, an iron index increase can be noted, as can a sharp drop in LILE concentration and fluorine content.

The concentrations of REEs and most HFSEs in late-phase leucogranites are relatively low and inconsistent. Zr concentrations vary from 12 to 62 ppm, Hf—0.54–2.8 ppm, Y—6.5–19 ppm, REE—90–121 ppm, and La/Yb$_N$ from 1.2 to 2.3, and negative Eu-anomalies also vary (Table S2, Figure 6). The REE distribution spectra of almost all samples show a pronounced M-type tetrad effect (T = 1.1–1.24). In contrast to the granite–leucogranites of the major phase, there is no significant correlation between silica with HFSE and REE content in late leucogranites.

Muscovite aplite dikes have $SiO_2$ at 74.8–75.55 wt.%, $K_2O/Na_2O$ at 1.14–1.52, high alumina content (A/CNK = 1.1–1.14), and a high iron index (*f:* 82–85%). They are enriched in LILEs, and depleted in Sr, Ba, and HFSE. Dikes show subsymmetric REE distribution spectra with La/Yb$_N$ = 0.8–1.5, Eu-minima, and a pronounced M-type tetrad effect (T = 1.18–1.2). In general, regarding their geochemical characteristics, aplites are similar to muscovite leucogranites of the late phase.

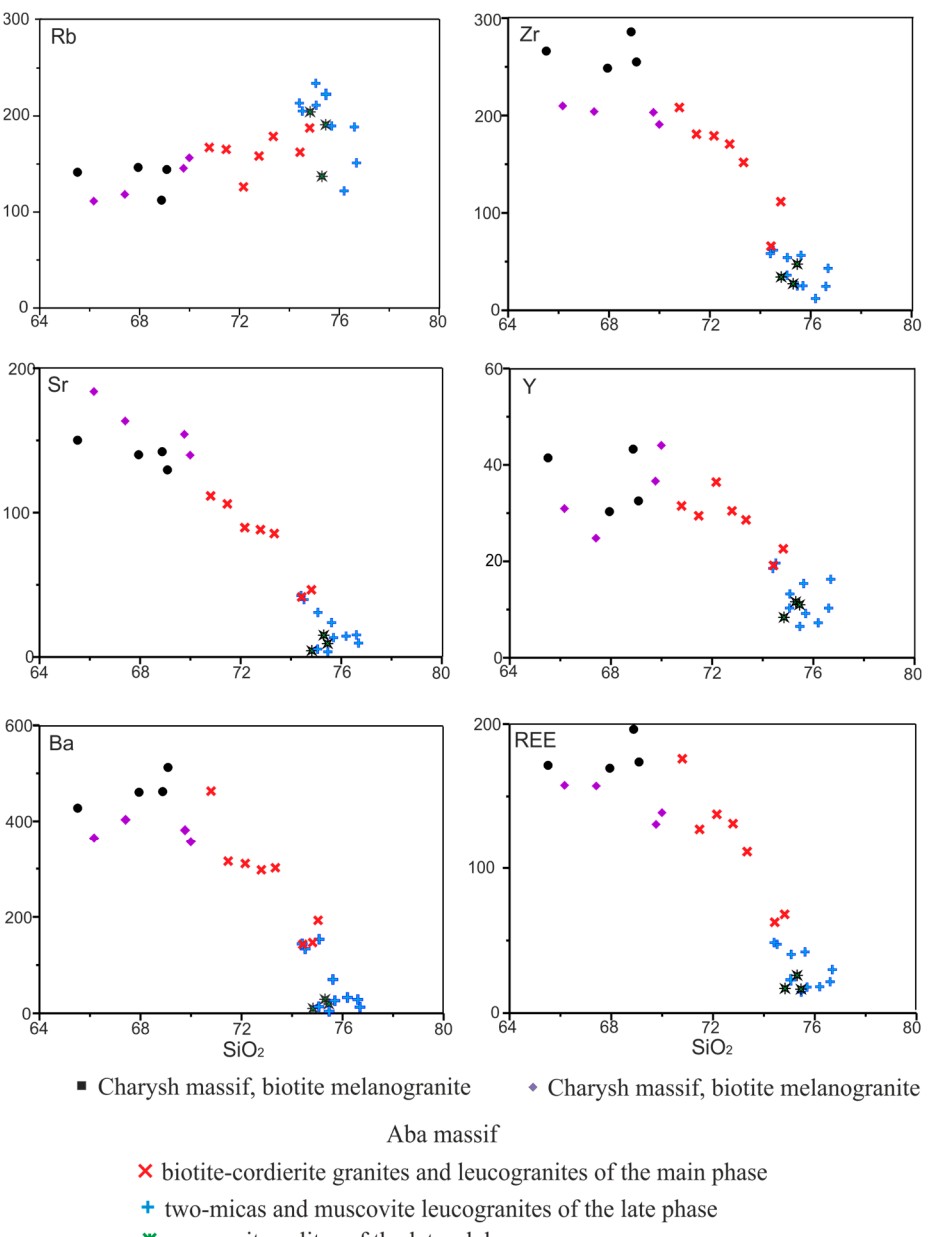

**Figure 7.** Binary geochemical diagram for Late Devonian granitoids of the western part of the Talitsa batholith (Gorny Altai).

The isotope characteristics of peraluminous granitoids in the western part of the Talitsa batholith are given in Table 1. Melanogranites of the Charysh intrusive have a positive value $\varepsilon Nd_{(T)}$ = +2, which distinguishes them from melanogranites of the Borovlyanka intrusion and granite–leucogranites of the Aba massif, which are characterized by near chondritic $\varepsilon Nd_{(T)}$. Oxygen isotope values of quartz show insignificant variations between the granite samples; values of $\delta^{18}O$ +15.0 and +15.4 were obtained for the melanocratic granites of the Charysh and the Borovlyanka intrusions, respectively. Biotite granites of the Aba major phase with low $SiO_2$ content are characterized by a high oxygen isotopic value in quartz ($\delta^{18}O$ +17.1) and an intermediate oxygen isotopic value in quartz from biotite–muscovite leucogranite ($\delta^{18}O \approx$ +16.3). For Aba granite–leucogranites, there is a gradual decrease in $\delta^{18}O$ values in minerals ranging from quartz to feldspar to biotite (Table 1). The observed sequence does not correspond to that of mineral crystallization, and reflects the closure temperatures of their isotope systems [45,46].

**Table 1.** Isotopic data for Late Devonian granitoids of the western part of the Talitsa batholith (Gorny Altai).

| № | Sp № | Nd ppm | Sm ppm | $^{147}Sm/^{144}Nd$ | $^{143}Nd/^{144}Nd$ | $\varepsilon_{Nd}(0)$ | $\varepsilon_{Nd}(t)$ | $T_{DM}$ | $T_{DM}$-2 | phase | $\delta^{18}O$ |
|---|---|---|---|---|---|---|---|---|---|---|---|
| 1 | 11-39 | 8.02 | 38.1 | 0.1272 | $0.512604 \pm 2$ | −0.7 | 2.7 | 965 | 921 | | |
| 2 | 12-56 | | | | | | | | | quartz | 15 |
| 3 | 04-27 | 5.18 | 24.67 | 0.1269 | $0.512453 \pm 9$ | −3.6 | −0.2 | 1226 | 1166 | | |
| 4 | 12-53 | | | | | | | | | quartz | 15.4 |
| 5 | 21-45/1 | | | | | | | | | quartz | 17.1 |
| 6 | 21-43/1 | | | | | | | | | feldspar | 15.5 |
| | | | | | | | | | | biotite | 11.1 |
| | | | | | | | | | | quartz | 16.5 |
| | | 5.3 | 24.1 | 0.1329 | $0.51249 \pm 10$ | −2.9 | 0.2 | 1247 | 1130 | feldspar | 15.7 |
| 7 | 21-41/2 | | | | | | | | | biotite | 12.7 |
| | | | | | | | | | | quartz | 16.28 |
| | | | | | | | | | | feldspar | 14.91 |
| | | | | | | | | | | biotite | 8.31 |
| 8 | 11-37/1 | 1.69 | 6.7 | 0.1520 | $0.512677 \pm 9$ | −2.1 | 0.0 | 1536 | 1144 | | |
| 9 | 21-9/3 | | | | | | | | | quartz | 16.3 |
| | | | | | | | | | | feldspar | 15.3 |
| | | | | | | | | | | muscovite | 14 |

Notes: The $\varepsilon_{Nd}(t)$ values are calculated for an age of 380 Ma.

### 4.4. Mineral Chemistry

#### 4.4.1. Mica

Dark micas in high-silica granitoids of the western part of the Talitsa batholith correspond to moderately ferruginous, high-alumina annite (Figure 8a). In biotites from melanocratic granites of the Borovlyanka massif, the content of $TiO_2$ is 2.7–3.7 wt.%, the iron index varies from 69 to 72%, and the alumina content is 50%–56%. The main impurities are $Na_2O$ (0.05–0.1 wt.%) and MnO (0.44–0.53 wt.%) (Table 2).

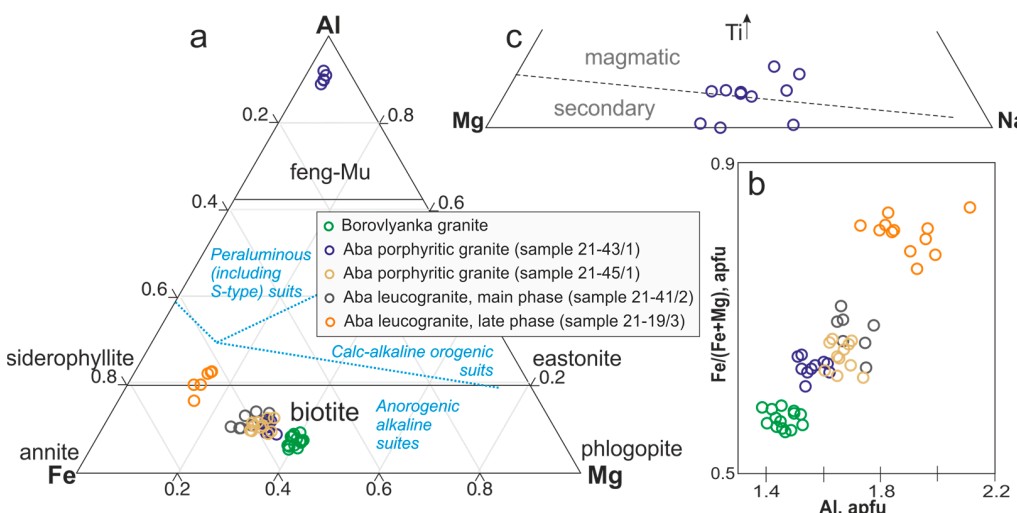

**Figure 8.** Compositions of analyzed micas. (**a**) Ternary diagram showing compositions of melanocratic micas with biotite endmembers; (**b**) Composition of biotite from Aba and Borovlyanka granites expressed in a Fe/(Fe + Mg) versus ΣAl diagram, also known as the annite (Ann)—siderophyllite (Sdp)—phlogopite (Phl)—eastonite (Eas) quadrilateral. Fe—total iron, i.e., $Fe^{2+} + Fe^{3+}$; (**c**) muscovites compositions in terms of Ti, Mg, and Na (atomic proportions).

**Table 2.** Electron microprobe analyses of micas of granitoids of the western part of the Talitsa batholith (Gorny Altai). Oxides in wt.%. bdl—below detection limit, n.d.—was not determined, $f$ = FeO$_{tot}$/(FeO$_{tot}$ + MgO). Structural formulae calculated on the basis of 11 oxygen atoms.

| Massif Rock Sample (Number of Measurement) | Aba | | | | Borovlyanka |
|---|---|---|---|---|---|
| | Porph.Granite (21-43/1) (10) | Porph.Granite (21-45/1) (11) | Leucogranite Main Phase (21-41/2) (7) | Leucogranite Late Phase (21-19/3) (5) | Granite (12-052) (14) |
| SiO$_2$ | 35.2–36.1 35.4 | 33.6–34.4 34.1 | 33.4–34.3 33.9 | 33.8–35.4 34.5 | 35.7–36.9 36.1 |
| TiO$_2$ | 3.17–4.56 3.84 | 3.19–4.34 3.65 | 1.56–3.66 2.84 | 1.92–2.42 2.17 | 2.7–3.7 3.3 |
| Al$_2$O$_3$ | 17.1–18.2 17.6 | 17.8–19.3 18.4 | 18.1–19.6 18.7 | 18.4–20.2 19.6 | 15.2–17.2 16.4 |
| MgO | 6.84–7.74 7.14 | 6.38–7.58 6.90 | 5.67–7.38 6.37 | 2.77–3.24 3.05 | 8.8–9.6 9.1 |
| FeO | 21.9–22.4 22.1 | 21.9–24.4 23.0 | 23.1–25.6 24.4 | 23.4–26.4 24.7 | 21.0–22.2 21.7 |
| MnO | 0.43–0.67 0.52 | 0.49–0.68 0.57 | 0.55–0.63 0.59 | 1.16–1.46 1.32 | 0.44–0.53 0.49 |
| CaO | bdl–0.12 0.06 | bdl–0.03 0.02 | bdl–0.31 0.13 | 0.03–0.36 0.14 | bdl–0.11 0.07 |
| Na$_2$O | 0.08–0.26 0.13 | 0.03–0.15 0.07 | bdl–0.08 0.05 | 0.02–0.09 0.06 | 0.05–0.10 0.07 |
| K$_2$O | 8.91–9.6 9.29 | 9.13–9.54 9.38 | 8.75–9.53 9.17 | 7.24–9.13 8.50 | 9.4–9.9 9.8 |
| Rb$_2$O | bdl–0.24 0.16 | bdl–0.18 0.13 | bdl–0.15 0.13 | 0.13–0.33 0.22 | bdl-0.11 0.11 |
| Cs$_2$O | bdl–0.04 0.04 | n.d. | n.d. | 0.05–0.13 0.08 | n.d. |
| BaO | bdl–0.33 0.23 | bdl–0.49 0.15 | bdl–0.17 0.11 | bdl–0.05 0.04 | bdl–0.18 0.11 |
| F | 0.65–0.92 0.80 | 0.58–0.78 0.71 | 0.56–0.80 0.65 | 0.56–0.73 0.64 | 0.37–0.59 0.49 |
| Cl | 0.04–0.07 0.05 | 0.02–0.03 0.02 | 0.02–0.05 0.03 | bdl–0.04 0.03 | 0.02–0.07 0.04 |
| Total | 96.6–98.3 97.2 | 96.6–97.4 97.0 | 95.8–97.7 97.0 | 93.7–96.2 95.0 | 95.9–98.6 97.5 |
| Formula units | | | | | |
| Si | 2.68–2.70 2.69 | 2.59–2.63 2.61 | 2.58–2.64 2.61 | 2.68–2.75 2.71 | 2.71–2.76 2.73 |
| Ti | 0.18–0.26 0.22 | 0.18–0.25 0.21 | 0.09–0.21 0.16 | 0.11–0.14 0.13 | 0.15–0.21 0.19 |
| Al | 1.53–1.62 1.57 | 1.61–1.74 1.66 | 1.65–1.78 1.70 | 1.73–1.85 1.81 | 1.39–1.53 1.46 |
| Mg | 0.78–0.88 0.81 | 0.73–0.87 0.79 | 0.65–0.84 0.73 | 0.33–0.38 0.36 | 0.99–1.08 1.03 |
| Fe | 1.39–1.43 1.40 | 1.41–1.57 1.47 | 1.48–1.65 1.57 | 1.52–1.75 1.63 | 1.33–1.43 1.37 |
| Mn | 0.03–0.04 0.03 | 0.03–0.04 0.04 | 0.04–0.04 0.04 | 0.08–0.10 0.09 | 0.03–0.03 0.03 |
| Ca | 0.001–0.009 0.005 | bdl–0.002 0.002 | bdl–0.026 0.011 | 0.003–0.031 0.012 | bdl–0.009 0.005 |
| Na | 0.01–0.04 0.02 | 0.00–0.02 0.01 | bdl–0.01 0.01 | 0.0–0.01 0.01 | 0.01–0.01 0.01 |
| K | 0.86–0.93 0.90 | 0.89–0.93 0.92 | 0.87–0.94 0.90 | 0.73–0.90 0.85 | 0.91–0.96 0.94 |
| Rb | bdl–0.01 0.01 | bdl–0.01 0.01 | bdl–0.01 0.01 | bdl–0.02 0.01 | bdl–0.01 0.01 |

Table 2. *Cont.*

| Massif<br>Rock<br>Sample<br>(Number of<br>Measurement) | Aba | | | | Borovlyanka |
|---|---|---|---|---|---|
| | Porph.Granite<br>(21-43/1)<br>(10) | Porph.Granite<br>(21-45/1)<br>(11) | Leucogranite<br>Main Phase<br>(21-41/2)<br>(7) | Leucogranite Late<br>Phase<br>(21-19/3)<br>(5) | Granite<br>(12-052)<br>(14) |
| Cs | bdl–0.001<br>0.001 | n.d. | n.d. | 0.002–0.004<br>0.003 | n.d. |
| Ba | bdl–0.010<br>0.007 | bdl–0.015<br>0.004 | bdl–0.005<br>0.003 | bdl–0.001<br>0.001 | bdl–0.005<br>0.003 |
| F | 0.16–0.22<br>0.19 | 0.14–0.19<br>0.17 | 0.14–0.19<br>0.16 | 0.14–0.18<br>0.16 | 0.09–0.14<br>0.12 |
| Cl | 0.01–0.01<br>0.01 | 0.00 | 0.00–0.01<br>0.00 | 0.00–0.01<br>0.00 | 0.00–0.01<br>0.01 |
| OH$^-$ | 1.77–1.84<br>1.80 | 1.81–1.85<br>1.82 | 1.80–1.86<br>1.84 | 1.81–1.85<br>1.84 | 1.86–1.90<br>1.88 |
| *f* | 0.74–0.76<br>0.76 | 0.75–0.79<br>0.77 | 0.76–0.82<br>0.79 | 0.88–00.90<br>0.89 | 0.69–0.72<br>0.70 |

The biotites of granites of the major phase of the Aba intrusive show wider compositional variation: the least silicic porphyritic granites (sample 21-45/1, $SiO_2$—70.79 wt.%) have higher $TiO_2$ than the melanogranites of the Borovlyanka massif ($TiO_2$—3.19–4.07 wt.%), $Na_2O$ (0.04–0.13 wt.%), and MnO (0.49–0.68 wt.%). The iron index of biotites is 74%–79%, and the alumina content is 57%–64%. In dark micas of more silicic granites (sample 21-43/1, $SiO_2$—72.78 wt.%), $Na_2O$ slightly increases and MnO decreases, but $TiO_2$, FeO, and $Al_2O_3$ do not change significantly (Table 2). In the leucogranites of the major phase (sample 21-41/2, $SiO_2$—74.82 wt.%), biotites are depleted in $TiO_2$ (1.56–3.66 wt.%) and $Na_2O$ (0.03–0.08 wt.%) and have a higher iron index (*f* = 76–82%). At the same time, the concentrations of MnO and $Al_2O_3$ in micas do not change at all (Table 2).

In biotites from melanogranites of the Borovlyanka intrusion, the fluorine content varies from 0.37 to 0.66 wt.% and chlorine reaches up to 0.07 wt.%. Meanwhile those from the porphyritic granites of the major phase of the Aba massif (sample 21-43/1) have higher concentrations of fluorine (0.58–0.78 wt.%), and chlorine content rarely reaches the detection limit (d.l.) of an X-ray microanalyzer. In more silicic granites of the Aba massif (sample 21-45/1), the fluorine content in biotites is the highest (F: 0.65–0.92 wt.%), and the chlorine content reaches 0.07 wt.%. Finally, in most of the analyzed biotites of the more silicic leucogranites (sample 21-41/2), the fluorine content does not exceed 0.6 wt.%, while in the central parts of large grains, it reaches 0.8 wt.%. Chlorine concentrations in these rocks are below the d.l. Biotites from two-mica leucogranites of the late phase are characterized by lower fluorine contents (F: 0.56–0.63 wt.%), and chlorine concentrations exceed the d.l. only in rare cases.

The "annite–siderophyllite–phlogopite–eastonite quadrilateral" (ASPE) diagram (Figure 8b) displays a regular trend in biotite compositional variation in granites of the Borovlyanka massif, and then, in two-mica granites of the Aba massif. Such a trend is consistent with strongly contaminated reduced I-type granites [47,48].

Two-mica granites of the Aba massif also contain light-colored micas (i.e., muscovites). The "Mg-Ti-Na" diagram [49] plots the composition of muscovites in a magmatic field (Figure 8c). The concentrations of isomorphic additives are recorded for FeO (1.67–2.29 wt.%), MgO (0.57–0.91 wt.%), and $Na_2O$ (0.50–0.76 wt.%), and those of $TiO_2$ (below 0.52 wt.%) and MnO (below 0.11 wt.%) are also noted. The content of fluorine is much lower than in biotites (F: 0.17- 0.42 wt.%), and chlorine does not exceed the d.l.

### 4.4.2. Feldspar

The chemical compositions of feldspars in granites and leucogranites are alike. K-feldspars are mainly represented by microcline with a twinning pattern. The amount

of orthoclase end-member in K-feldspar varies between 92 and 97% in biotite granites from the major phase intrusions, and from 95 to 96% in the late-phase fine-grained two-mica granites (Figure 9). The composition of K-feldspars is close to stoichiometric, with a constant isomorphic admixture of $Na_2O$ (0.36–0.93 wt.%) and BaO (0.03–0.33 wt.%), and in rare cases, MnO (up to 0.02 wt.%), CaO (up to 0.05 wt.%), $Rb_2O$ (up to 0.12 wt.%), and $Cs_2O$ (up to 0.03 wt.%) (Table 3).

**Table 3.** Electron microprobe analyses of feldspars of the Aba massif.

| Rock Sample Mineral (Number of Measurement) | Porph.Granite (21-43/1) | | Leucogranite Late Phase (21-19/3) | |
|---|---|---|---|---|
| | KFsp (7) | Pl (8) | KFsp (2) | Pl (8) |
| $SiO_2$ | 63.7–65.0<br>64.6 | 63.3–68.4<br>67.0 | 63.2–65.1<br>64.1 | 64.6–68.8<br>66.5 |
| $Al_2O_3$ | 18.0–18.6<br>18.4 | 19.4–22.4<br>20.3 | 18.5–18.6<br>18.5 | 19.5–22.2<br>20.7 |
| FeO | bdl–0.05<br>0.03 | bdl–0.07<br>0.05 | 0.02–0.03<br>0.02 | bdl–0.04<br>0.03 |
| MnO | bdl–0.01<br>0.01 | bdl–0.07<br>0.05 | bdl–0.02<br>0.02 | bdl–0.02<br>0.02 |
| CaO | bdl–0.04<br>0.03 | 0.14–3.44<br>0.84 | 0.03–0.05<br>0.04 | 0.09–2.71<br>1.34 |
| $Na_2O$ | 0.36–0.93<br>0.58 | 10.0–12.1<br>11.5 | 0.45–0.52<br>0.48 | 9.9–11.9<br>11.1 |
| $K_2O$ | 16.2–17.5<br>16.9 | 0.05–0.13<br>0.08 | 16.6–17.1<br>16.8 | 0.04–0.65<br>0.23 |
| $Rb_2O$ | bdl–0.07<br>0.07 | bdl–0.07<br>0.07 | bdl–0.12<br>0.12 | bdl |
| $Cs_2O$ | bdl–0.03<br>0.03 | bdl | bdl–0.02<br>0.02 | bdl–0.04<br>0.03 |
| SrO | | | | |
| bdl | 0.03–0.33<br>0.15 | bdl-0.05<br>0.03 | bdl | bdl-0.04<br>0.03 |
| Total | 100.2–101.0<br>100.7 | 99.3–100.5<br>99.8 | 99.5–100.9<br>100.2 | 99.4–100.4<br>99.9 |
| Formula units | | | | |
| Si | 2.96–3.00<br>2.99 | 2.82–2.99<br>2.95 | 2.96–2.99<br>2.98 | 2.85–3.00<br>2.92 |
| Al | 0.99–1.02<br>1.00 | 1.01–1.17<br>1.05 | 1.00–1.03<br>1.01 | 1.00–1.15<br>1.07 |
| Fe | 0.001–0.002<br>0.001 | 0.001–0.002<br>0.001 | 0.001–0.001<br>0.001 | bdl–0.002<br>0.001 |
| Mn | bdl–0.000<br>0.000 | bdl–0.001<br>0.001 | bdl–0.001<br>0.001 | bdl–0.001<br>0.001 |
| Ca | bdl–0.000<br>0.000 | 0.01–0.16<br>0.04 | 0.00–0.00<br>0.00 | 0.00–0.13<br>0.06 |
| Na | 0.03–0.08<br>0.05 | 0.86–1.02<br>0.98 | 0.04–0.05<br>0.04 | 0.85–1.01<br>0.95 |
| K | 0.95–1.04<br>0.99 | 0.003–0.007<br>0.005 | 1.02–0.97<br>0.99 | 0.002–0.037<br>0.013 |
| Rb | bdl–0.002<br>0.002 | bdl–0.002<br>0.002 | bdl–0.004<br>0.004 | bdl |
| Cs | bdl–0.001<br>0.001 | bdl | bdl–0.000<br>0.000 | bdl–0.001<br>0.001 |
| Sr | | | | |
| bdl | 0.001–0.006<br>0.003 | bdl-0.001<br>0.001 | bdl | bdl-0.001<br>0.001 |
| Ab | 0.031–0.080<br>0.050 | 0.835–0.990<br>0.956 | 0.039–0.045<br>0.042 | 0.839–0.992<br>0.926 |
| An | 0.000–0.002<br>0.001 | 0.007–0.158<br>0.039 | 0.002–0.003<br>0.002 | 0.004–0.127<br>0.062 |
| Or | 0.920–0.969<br>0.950 | 0.003–0.007<br>0.005 | 0.953–0.959<br>0.956 | 0.002–0.036<br>0.013 |

Oxides in wt.%. bdl—below detection limit, n.d.—was not determined. Structural formulae calculated on the basis of 8 oxygen atoms.

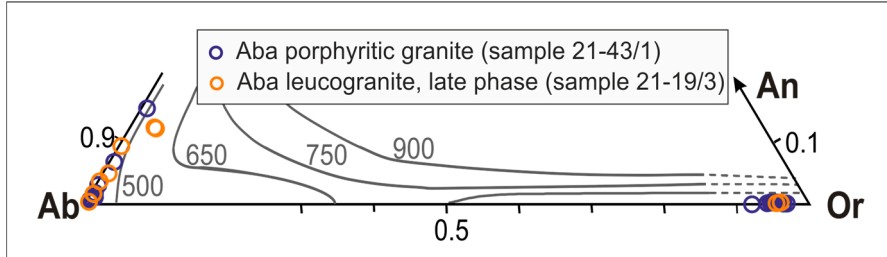

**Figure 9.** Ternary diagram showing feldspar compositions with Ab-An-Or endmembers.

Plagioclase often undergoes a secondary alteration (saussuritization). The compositions of plagioclases from biotites and two-mica granites are similar (Figure 9). The content of albite end-member varies from 83.5 to 99.2% ($K_2O$: 0.04–0.65 wt.%) with low impurity content (MnO: less than 0.02 wt.%, BaO: 0.05 wt.%). Plagioclases from two-mica granites also have $Cs_2O$ (up to 0.04 wt.%) and SrO (up to 0.16 wt.%) (Table 3).

### 4.5. Quartz-Hosted Parental Magma Fluid and Melt Inclusions

To determine the fluid-present phase of the granitoid evolution of the Aba massif, we examined the inclusion associations of the mineral-forming media in quartz from medium–coarse-grained biotite granite–leucogranites of the major phase (samples 21-42/1 and 21-41/2), and from two-mica and muscovite fine–medium-grained leucogranites of the late phase; we also studied late-phase pegmatoid segregations (samples 21-20/2, 21-19/1, 21-19/4, and 21-19/6). Because the size of inclusions, as a rule, are not larger than several micrometers we were not able to obtain precise data on fluid density and fluid and melt chemical compositions. Thus, discussion about the role of fluids in this study is based solely on petrographic observations and should be considered as preliminary and qualitative.

Earlier, we discussed that the granite–leucogranites of the major phase and leucogranites of the late phase have signs of brittle-plastic deformations. Deformation processes caused the formation of mechanical twins, dislocation boundaries, recrystallization, and granulation of the magmatic quartz. As a result, the inclusions rearrange according to the mechanism described by [50,51]. For this reason, only inclusions from relics of undeformed cores of quartz grains can be used to characterize the process of magmatic crystallization.

In sample 21-41/2—granite–leucogranite of the major phase of the Aba massif with signs of brittle–ductile deformations—essentially gaseous fluid inclusions are confined to stress-related fractures and grain boundaries in the quartz mosaic. In the undeformed parts of the grains, we observe azonal groups of large fluid inclusions (Figure 10a). The volume fraction of the gas bubble in them does not exceed 15%, in contrast to the inclusions observed for quartz in two-mica leucogranites.

Other groups of inclusions measuring no larger than 2 μm were also found in the quartz of this sample. Both groups of inclusions are visible only under high magnification. This fluid inclusion assemblage is dominated by fluid, two-phase liquid–vapor inclusions, but acicular crystalline inclusions, single plates, and intergrowths of muscovite are also present. Quartz from granite–leucogranite (sample 21-42/1) is practically uninfluenced by the deformation. Groups of larger inclusions mainly consist of fluid, two-phase liquid–vapor inclusions of about 5–20 μm in size. The volume fraction of the gas phase in them is 5%–10%. Some inclusions contain crystalline phases, but their nature requires clarification. The smaller inclusions are associated with mineral phase inclusions, which can either be transparent or opaque acicular crystals. These groups contain what are interpreted to be syngenetic fluid and melt inclusions (Figure 10b).

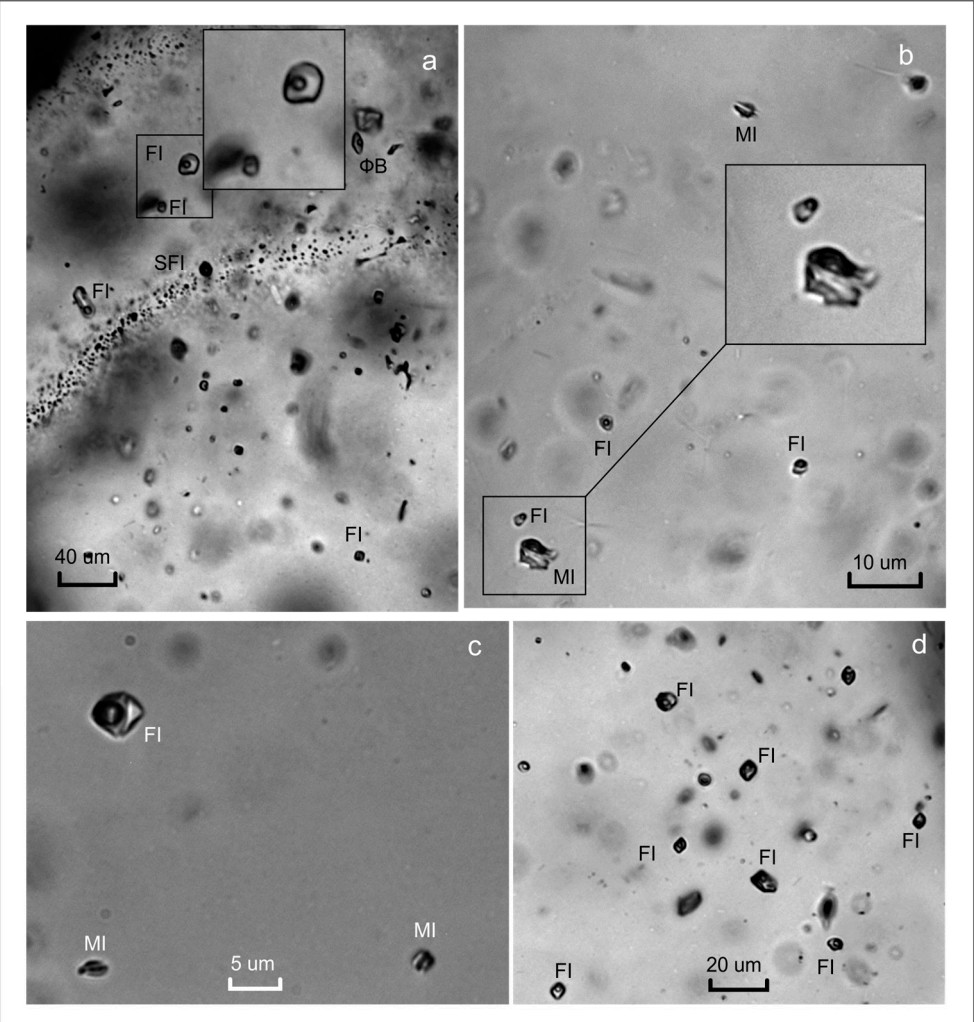

**Figure 10.** Quartz-hosted fluid and melt inclusions from Aba massif: (**a**) irregularly distributed primary two-phase fluid inclusions and the healed fracture with secondary gas-rich fluid inclusions (SFI) (21-41/2); (**b**) the assemblage of co-eval melt and fluid inclusions (21-42/1); (**c**) primary melt inclusions (MI) accompanying the single two-phase fluid inclusion (FI) (21-19/6); (**d**) assemblage of primary two-phase fluid inclusions irregularly distributed in the host quartz (21-20/1).

The two-mica leucogranites from the late phase of the Aba massif were subjected to a large degree of deformation compared to the granite–leucogranites of the major phase. In this regard, the search for associations of inclusions in them was difficult.

The most informative sample is 21-19/6, which contains a small pegmatoid segregation with large quartz crystals. In comparison to the host granite, the segregation is practically uninfluenced by the deformation, and its quartz contains groups of what are interpreted to be primary and secondary inclusions of the parental magma. Primary inclusions are extremely small, rarely reaching 10 µm in size, and they usually form azonal groups. They are more often found in the marginal parts of the grains, and less often in the central zones of pegmatoid quartz. The fluid inclusions are in two-phase liquid–vapor form. The volume fraction of the vapor phase is 20%. Sometimes, crystallized melt inclusions occur near fluid inclusions (Figure 10c).

In leucogranites, primary inclusions are extremely rare. More often, these inclusions are confined to stress-related fractures, have sinusoidal configurations, or form groups along grain boundaries in a mosaic quartz. Moreover, the inclusions are mainly in liquid–vapor form.

The medium-grained two-mica leucogranite sample 21-20/2 is the least deformed. Quartz granulation is noticeable along the marginal parts of the grains and very rarely within them. We observed azonal groups of inclusions only in undeformed quartz cores, both in the center and on the periphery of the grains. The inclusions (no larger than 5 μm in size) are in fluid, two-phase liquid–vapor form, with a gas phase volume fraction of about 20%. (Figure 10d). The small inclusions in this sample form azonal groups, which, in addition to typical fluid inclusions, are represented by microcrystals (zircon, an unidentified acicular mineral) and rare crystallized melt inclusions.

## 5. Discussion

The investigated granitoids of the western part of the Talitsa batholith have been classified as S-types, formed through the partial melting of metasedimentary source rocks. Our previous studies [27,52] showed that high-alumina melanogranites, similar to the rocks of the Charysh and Borovlyanka intrusions, were formed as a result of partial melting of a combined source, including metasedimentary and metabasaltic rocks. This model source fully corresponds in composition to the Cambrian–Ordovician terrigenous–volcanogenic–siliceous strata of the Zasurya Group of Western Altai [53–55]. The observed variations in Nd isotope composition in the granitoids of the Charysh and Borovlyanka intrusions are explained by contributions from different proportions of metasedimentary and metavolcanic rocks. Additionally, Nd isotope data indicate the same source of formation for melanocratic granites of the Borovlyanka intrusion and for leucocratic granites of the Aba massif.

The results of our geochronological studies suggest a short formation timescale for all the studied granitoids (no more than approximately five million years). Therefore, we have reason to believe that the formation of Aba leucocratic granites and rocks of the Charysh and Borovlyanka massifs are associated with the evolution of the same magma, rather than the remelting of previously formed granitoids. The least siliceous varieties of granite of the major phase of the Aba massif are quite similar, in terms of their petrographic characteristics, mineralogy, and chemical composition, to the melanogranites of the Borovlyanka intrusion, and it is likely that these rocks are products of the evolution of a single batch of magma.

Magma, which was the source for the granite–leucogranites of the major phase of the Aba massif, evolved in the same way as a typical granite system. This is expressed as an increase in silica with an approach toward the composition of the surface granite cotectic, an increase in the contents of LILEs and volatile elements, and a decrease in the concentrations of alkaline Earth metals. Accessory minerals, such as monazite, xenotime, possibly zircon, and cheralite had a significant impact on the process of decreasing the ratio of HFSEs, REEs (mainly LREEs), and La/Yb$_N$. At the same time, the slightly negative Eu-anomaly demonstrates a minor role of feldspar fractionation in this process. The presence of facies contacts between the granites and leucogranites of the major phase of the Aba massif, as well as the much higher σ$^{18}$O values compared to other granitoids of the western part of the Talitsa batholith, indicate that differentiation occurred in a shallow chamber at the level of intrusion formation, with the active participation of crustal fluid.

In quartz of the leucogranites of the major phase, we found what we interpret to be small syngenetic fluid and melt inclusions. Although the size of the inclusions hindered our ability to study them in detail, the mere presence of such inclusions suggests the exsolution of a fluid phase (volatile phase) from the volatile saturated melt during the final stage of evolution. The azonal characterization of associations consisting only of fluid inclusions suggests that they are likely the result of magmatic fluid entrapment during magma evolution. The primary nature of such inclusions is confirmed by their association with crystalline inclusions that became trapped during quartz growth, and which do not have a secondary nature. Certainly, the tectonic deformations that affected the granitoids caused quartz recrystallization, the transformation of primary inclusions, and the origin of secondary ones. However, the characteristics of undisturbed primary associations support our findings.

The appearance of what appear to be syngenetic melt and fluid inclusions in quartz from the leucogranites of the major phase corresponds with a slight decrease in fluorine content (both in biotite and in the whole rock), and to the occurrence of M-type tetrad effects in the REE distribution spectra (Figure 6). Thus, the $K_2O/Na_2O$ ratio in leucogranites is not below 1.5, and the LILE concentrations are the same as, or even exceed, those in the least silicic rocks; this can be explained by the small fluid fraction in the magma.

A different situation is observed for the two-mica and muscovite granites of the late phase and aplite dikes. The depletion of HFSEs and REEs and the quasi-symmetric REE distribution spectra with negative anomalies are compatible with a significantly higher degree of melt differentiation during the formation of these rocks. The presence of intrusive contacts with granite–leucogranites of the major phase, and the lower $\sigma^{18}O$ values, indicate that their evolution occurred in a chamber located at a depth below that of intrusion formation. In addition, all the studied rocks have significant M-type tetrad effects on the REE distribution spectra, and our study of quartz phenocrysts indicates high fluid saturation and a compositionally heterogeneous mantle source, even for the lowest-silica rocks (sample 21-20/1). The content of fluorine in both rocks and micas of the late-phase leucogranites is minimal, and an increase in $SiO_2$ will cause a decrease in alkalinity, a decrease in $K_2O/Na_2O$ ratios (to 0.77–0.92 in the most siliceous varieties), and a decrease in LILE concentrations (Figure 7a). All the characteristics given above, as well as the association of what are interpreted to be syngenetic fluid and melt inclusions in the undeformed quartz cores of these rocks, and pegmatoid segregations, indicate a more significant role of a fluid phase in the magma during the formation of the granite–leucogranites of the major phase. A detailed petrographic study of rocks in thin sections allows us to assert that leucogranite magma degassing did not cause metasomatic transformation (for example, the formation of late albite). In summary, the geochemical characteristics and mineralogical composition described above are determined by magma evolution and interaction with a fluid phase.

## 6. Conclusions

Based on the research conducted herein, we arrived at the conclusion that the formation of high-silica magmas, from which the leucogranites of the Aba massif crystallized, are associated with the separation of a fluid phase and the degassing of fluid-saturated granitoid magmas. At the same time, for granite–leucogranites of the major-phase and late-phase leucogranites, this process occurred differently.

Magma evolution, from which the granite–leucogranites of the major phase originated, took place in shallow magma chambers (at the level of intrusion formation). The exsolution of a fluid phase (volatile phase) from the volatile saturated melt occurred at a late stage of crystallization. The amount of free fluid was small, the system remained almost closed, and degassing was weak. As a result, the tetrad effect and depletion in fluorine were extremely low. In general, we note only a slight change in the composition of the melts and the rocks crystallized from them. Silica-enriched rocks (with a large amount of silica compared to the composition of the granite cotectic) are not widespread. Their geochemical characteristics confirm their formation with silica input through a fluid phase, but the resulting magmas contained minimal excess silica.

The parent melt of the late-phase leucogranites evolved in a deeper chamber. The process of differentiation occurred at greater depth and was more prolonged (the shallow chamber had most likely already crystallized). The presence of what are interpreted to be syngenetic melt and fluid inclusions indicate that the leucogranitic magma, which is close in composition to the granite minimum, was already saturated with volatiles in the early crystallization stages. Then, when quartz phenocrysts had already formed, migration of the fluid phase outside the chamber occurred during the opening of the system, which led to a decrease in fluorine and some lithophile elements (including K and Rb). The resulting magmas were silica-enriched and contain a large amount of normative quartz. We assume that partial degassing occurred simultaneously with the intrusion of leucogranite

magma into the upper crustal horizons (at the level of intrusion formation). During the ascent, the magma became depleted in volatile elements and quickly crystallized, leading to the formation of fine–medium-grained rock textures and the development of brittle-plastic deformation textures, which can be considered indicators of growing viscosity of the intruding magma. In places of fluid phase localization, which were retained inside the crystallized massif in the late stages of leucogranite magma crystallization, pegmatoid schlieren was formed without any sign of tectonic deformation.

Our studies have shown that the high-alumina granitoids of the Aba massif (leucogranites with a silica content that exceeds the composition of the granitic minimum) did not originate from the accumulation of silica during fractional crystallization, but are related to the volatile saturation of magmatic melts, the exsolution of a fluid phase, and magma degassing. The most effective mechanism for the formation of silica-enriched rocks is the removal of large-ion lithophilic components, primarily potassium, while a silica transfer process leads to the formation of magmas with minimal excess silica.

**Supplementary Materials:** The following supporting information can be downloaded at: https://www.mdpi.com/article/10.3390/min13040496/s1, Table S1: Zircon U-Th-Pb isotopic data &calculated ages for Late Devonian granitoids of Western part of the Talitsa batholith; Table S2: Major (wt.%) and trace (ppm) element compositions for Late Devonian granitoids of the western part of the Talitsa batholith (Gorny Altai).

**Author Contributions:** Conceptualization, N.N.K. and S.Z.S.; formal analysis, O.A.G., E.A.K. and D.V.S.; investigation, N.N.K., O.A.G., S.Z.S., E.A.K., S.N.R. and D.V.S.; writing—original draft preparation, N.N.K., O.A.G., S.Z.S. and S.N.R.; writing—review and editing, O.A.G.; visualization, N.N.K., O.A.G., S.Z.S. and E.A.K.; supervision, N.N.K.; project administration, N.N.K.; funding acquisition, N.N.K. All authors have read and agreed to the published version of the manuscript.

**Funding:** The study was financially supported by the Russian Science Foundation (project 21-07-00175).

**Data Availability Statement:** The data presented in this study are available on request from the corresponding author.

**Acknowledgments:** The authors are grateful to referees for constructive criticism and suggestions that led us to improve the text of the manuscript and the presentation of the results.

**Conflicts of Interest:** The authors declare no conflict of interest.

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
