# Peer review of "Formation of High-Silica Leucocratic Granitoids on the Late Devonian Peraluminous Series of the Russian Altai: Mineralogical, Geochemical, and Isotope Reconstructions"

_minerals, doi:10.3390/min13040496_

Round 1

Reviewer 1 Report

Manuscript ID: Minerals-2092895-peer-review-v1

Title: Formation of high-silica leucocratic granitoids on the Late Devonian peraluminous series of the Russian Altai: mineralogical, geochemical, and isotope reconstructions

Authors: Nikolay N. Kruk, Olga A. Gavryushkina, Sergey Z. Smirnov,  Elena A. Kruk, Sergey N. Rudnev, Dina V. Semenova, Sergey V. Khromykh

This manuscript presents a variety of geological aspects including petrography, whole rock geochemistry, mineral chemistry, stable and unstable isotopes, fluid inclusion studies and geochronology of leucocratic granitoids of Russian Altai. The quality and quantity of data are optimum. I did not find any field photos and petrographic photomicrographs in the manuscript. Before introducing geochemistry and geochronology of rock suit, it is essential that some field and microscopic pictures to be presented. It is better that geochronology section to be put after geochemistry and mineral chemistry before Discussion section.

It is not clearly discussed in the manuscript that why some samples are ferroan and other ones are magnesian in Frost et al. (2001) diagram (Figure 4d).

I suggest that the mineral chemical data to be used more in the interpretation of petrogenesis and tectonic discrimination of the studied granitoids, for example considering the mineral chemistry of biotite, Abdel-Rahman (1994) diagrams can be drawn for distinguishing tectonic setting of the studied rocks.

Abdel-Rahman, A. F. M. (1994). Nature of biotite from alkaline, calc-alkaline, and peraluminous magmas. Journal of Petrology, 35(2), 525-541.

The whole part of the text, and figure and table captions should be checked for grammar and spell of words carefully.

After resolving above mentioned deficiencies the manuscript can be accepted for publication in the Minerals journal.

Author Response

We thank Referee for the constructive comments, which helped us to improve our manuscript. All of the comments have been addressed. Please see the attachment.

Reviewer 2 Report

Comments:

1.    The authors should describe in the abstract how this research is different from previous works carried out in the region. Also, add a paragraph on the significance of this research in the Introduction.

2.    Figures 1 and 2. The authors have used the term “addition”. Please describe the addition. Is it modification/addition of features in the figure?

3.    Rearrange the headings in the Analytical Methods as Mineral analyses, U-Pb Geochronology, Whole-rock geochemical analyses, Whole-rock isotopic analyses

4.    Please shift Petrograpy to 4.1 in the Results section followed by Composition of minerals, Geochronology, Geochemistry and isotope characteristics of rocks, to keep consistency in Analytical Methods and Results.

5.    Please rewrite conclusions highlighting the main points concluded from this research. Right now the conclusions part looks more like a discussion section.

6.    The English language of the paper needs a lot of improvement.

Author Response

We thank Referee for the constructive comments, which helped us to improve our manuscript. All of the comments have been addressed, please see the attachment.

Reviewer 3 Report

Line 13; What kind of minerals?

Line 15: What type of geological data and what type of isotope?

The abstract has to improve to make it more attractive to the future reader

Lines 26-27: references?

Line 30: [2 etc] ??

Lines 35-37: This paragraph needs reference

The first paragraph has a different spacing than the second paragraph, it must be consistent throughout.

Lines 52 and 53: Change the SiO2 and put the number 2 as a subscript

Lines 80 to 93: Is this description from your field work? If so, you do not need references, but if they are not your observations, you must cite the references of the authors who described the different types of rocks or formations

Lines 95 to 113: Similar to the previous paragraph, the geological description lacks references, this section needs to improve

Line 116: Where the samples were taken from, "localities", how many samples were taken. How many samples were sent for analysis of major and trace elements?

Line 120: What does IGM SB RAS mean?

Line 122: What is the standard method?

Lines 125-129: What were the parameters for fluoride iones analyses?

This section is not well understood. I think it's messy.

It does not mention how many samples are analyzed with XRF, ICP-MS, nor by fluoride ions

The ICP-MS section I do not understand what kind of samples the analyzes were performed on, and it should also mention the parameters used. I understand that you mention that you followed the protocols of [31] but some readers need to know what your parameters were used, what type of isotopes you measured, etc.

Line 135: what was the pH of the solution?

Line 142: What does E vs pF mean?

Lines 146 -161: In this section you have to improve all the subscript and superscript.

How many samples did you measure?

Line 166: EMPA means Electron Microprobe Analysis

What were the target minerals?

What elements were analyzed, what was the configuration of the crystals, what type of crystals were used?

What were the parameter used in scanning electron microscope?

What kind of minerals were measured by oxygen isotope composition?

Line 195: CL = cato luminescence?

How many zircon grains were dated?

Do you have any CL zircon figure?

This part should go in the methods section “(sample 12-056, coordinates 207 51o12’35.7”N; 83o48’38.2”E)”

Figure 3 has very small numbers, the headers cannot be read, the concord ages cannot be read either. The closed caption in figure 3 should change the name. U/Pb isotopic diagrams are generally not used

The age of 380 ± 3 Ma  does not match with that of figure 3a 379 +- 3.4 Ma

This part should go in the methods section “ (sample 12-053, coordinates 51o24’04.5”N; 83o49’13.7”E)”

Line 225: replace The grain size of zircons for “Zircons grain size”

Line 225: add μm  to 20 to 160

 Line 227: How can you identify by CL that the zonation of the zircons are magmatic? What difference would it have from a hydrothermal zonation?

It would be nice to see some zircons CL images. You mention a lot that the zoning is magmatic, but there are no images to prove these assertions

Line 234: Change “Age of rocks” for Rock Ages

Lines 234-247: This paragraph needs to improve. You could generate a table or an annex and place the samples and their locations

Line241: What do you mean by Kelong?

Line 242: Change 33 to thirty-three

Line 252 to 259: Where did these zircons come from? What is your location? What use would they serve me in the study that you are doing?

Line 263: add simbol ± (380 + 5 Ma).

You should include a figure with photos of thin sections for a better explanation of the petrographic section.

Line 279: How do you know that it is an acid plagioclase, what name would you give to this mineral? Any image to prove it?

Line 318: You have to mention the figures in order, first a, then b, etc.

Line 333: What do you mean by Femic?

Line 363: The figures must appear in order. move figure 4f

In figure 6 the units of the axes are missing

This is confusing “Legend is same as in |Fig. 3. “

Figure 7 close caption: With what method did they analyze the micas?

Line 531: How much would it be for you with high magnification? Figure 9?

Discussion section numeral should be 5

Missing acknowledge you section

You speak about a fluid inclusions study, but no data on pressure, temperature, etc. are observed.

Author Response

We thank Referee for the detailed review and constructive comments, which helped us to improve our manuscript. The most of the comments have been addressed. Please see the attachment.

Reviewer 4 Report

Manuscript ID: minerals-2092895

Title: Formation of high-silica leucocratic granitoids on the Late Devonian peraluminous series of the Russian Altai: mineralogical, geochemical, and isotope reconstructions

Dear Editor and authors,

I have read with interest the work of Kruk et al., where the authors present new original petrographic, whole-rock geochemistry, geochronological and isotopic data for two slightly peraluminous Late-Devonian intrusions/plutons that are part of the Talitsa batholith. The data presented in the manuscript are of very good quality and the authors provide new information on the studied rocks, giving important implications for the nature, genesis, and evolution of the investigated multi-stage granitoid magmatism. I believe the manuscript presents new interesting data that can bring important insights into the genesis and evolution of high-silica leucogranite rocks. However, the present text of the manuscript has various problems (including language, formatting, organization, and conceptual/editorial problems) and, in my view, should only be considered for publication after a major review.

Regarding the English language, I have made some recommendations in the annotated PDF file, but it would not be sufficient, and I strongly suggest a native speaker review of the text. Also, throughout the manuscript, there are several text inconsistences that must be corrected by the authors, including single phrase paragraphs that can easily be incorporated in the previous paragraph (please see some indications in the PDF). Authors have also to revise the text, using correctly the subscript and superscript formatting were necessary. In the following topics, I will provide some general comments regarding each section presented in this manuscript, but please refer to the PDF file for more detailed comments.

Abstract: The abstract is not elucidative of what will be described in the main manuscript and is also hard to follow. In the title of the manuscript, the authors highlight the high-silica content of the granitoid, but in the abstract, they refer to the investigated rocks as high-alumina. Also, the abstract should present a summary of the main results, which is lacking in this current version.

Introduction: The introduction presents well the scientific justification for the manuscript.

Geological Setting: This section is too simplified, which make the reader that are not familiar with the geology of this region a bit lost. For example, in the second paragraph of this section, it is stated that the Late-Devonian granitoid rocks are related to the change from subduction to a transform geodynamic setting; what is the orogenic system associated with these rocks? Also, in the figures (geological maps) from the area, there are several volcanic and sedimentary sequences that could be better addressed since the investigated Late-Devonian granitoid rocks are interpreted to be S-type granites. I think this is the place to present to the reader the main sedimentary sequences that could be the source of the investigated rocks together with their isotopic signature, etc. For further comments on this topic, please refer to the PDF file.

Analytical Techniques: Please see the annotations on the PDF file for some minor comments. My major problem is not with the techniques itself, but with the fact that the manuscript does not provide the tables with the analytical results for the zircon U-Pb dating (nor for the sample or the zircon standards used) or the lithogeochemistry. These tables must be included in the future version of the manuscript, either as within the manuscript or as a supplementary file.

Results: My major comments about the results topic are:

-        It would be better to start with the petrographic section. Also, why the authors did not show any figure or illustration of the outcrop pattern and thin-sections for the investigated rocks; as a work focused on the petrological evolution of granitoid rocks, so it would be nice to see the main aspects of the rocks.  

-        The geochemical section is too long, repetitive, and because of it, hard to follow. I think the authors can easily reduce this item, making it simpler and more concise!

-        Other minor comments and suggestions are highlighted in the annotated PDF file.

Discussion: Although I do not disagree with some interpretations, I think that the authors do not properly discuss their data. The way this section is now, there are several interpretations that are not anchored in the presented data. Please, see in the marked PDF the comments I have made for this section. In general, the discussion topic should be fully reorganized.

I apologize for any misunderstanding I may have occurred while reading the manuscript and wish the comments help to improve the manuscript.

Best regards.

Author Response

Dear Referee and Editor,

We are very grateful to the Referee for the detailed and helpful review and constructive comments, which helped us to improve our manuscript. We carefully studied the reviewer's notes (also in the pdf file) and the most of the comments have been addressed. Please see the attachment.

Round 2

Reviewer 2 Report

Comments:

1.    The authors have incorporated the comments and the paper is very much improved.

2.    Only minor English language corrections are needed.

Author Response

We thank Referee for the review. English language was corrected.

Author Response

We thank Referee for the detailed review and comments, which helped us to improve our manuscript. The most of the comments have been addressed. 

Reviewer 4 Report

Manuscript ID: minerals-2092895

Title: Formation of high-silica leucocratic granitoids on the Late Devonian peraluminous series of the Russian Altai: mineralogical, geochemical, and isotope reconstructions

Dear Editor and authors,

Most of the suggestions and appointments made by me in the first version of the manuscript have been addressed by the authors. In my view, the manuscript should be considered for publication after minor modifications. Although the ms improved in this version, it still contains some semantical and grammatical mistakes (mainly on the topic related to the presentation of geochemical data) and, thus, needs to be carefully proofread by a native speaker. I have made some minor suggestions in the annotated PDF, but I do not think it would be enough. However, I think the decision regarding the English quality can be addressed by the Minerals editorial board.

Best regards.

Author Response

Dear Referee and Editor,

We are very grateful to the Referee for the detailed and helpful review and constructive comments, which helped us to improve our manuscript. We corrected English language.